# The Free Energy Principle for Perception and Action: A Deep Learning Perspective

**DOI:** 10.3390/e24020301

**Published:** 2022-02-21

**Authors:** Pietro Mazzaglia, Tim Verbelen, Ozan Çatal, Bart Dhoedt

**Affiliations:** IDLab, Ghent University, 9052 Gent, Belgium; tim.verbelen@ugent.be (T.V.); ozan.catal@ugent.be (O.Ç.); bart.dhoedt@ugent.be (B.D.)

**Keywords:** free energy principle, active inference, deep learning, machine learning

## Abstract

The free energy principle, and its corollary active inference, constitute a bio-inspired theory that assumes biological agents act to remain in a restricted set of preferred states of the world, i.e., they minimize their free energy. Under this principle, biological agents learn a generative model of the world and plan actions in the future that will maintain the agent in an homeostatic state that satisfies its preferences. This framework lends itself to being realized in silico, as it comprehends important aspects that make it computationally affordable, such as variational inference and amortized planning. In this work, we investigate the tool of deep learning to design and realize artificial agents based on active inference, presenting a deep-learning oriented presentation of the free energy principle, surveying works that are relevant in both machine learning and active inference areas, and discussing the design choices that are involved in the implementation process. This manuscript probes newer perspectives for the active inference framework, grounding its theoretical aspects into more pragmatic affairs, offering a practical guide to active inference newcomers and a starting point for deep learning practitioners that would like to investigate implementations of the free energy principle.

## 1. Introduction

Understanding the processes that sentient and reasoning beings play out mentally in order to perceive the world they live and act in is as compelling as it is complex. The free energy principle hypothesizes all brain processes can be understood as subserving one unicum imperative: the minimization of free energy [1,2]. This principle, and its corollary active inference, assumes that agents act to contrast forces from the environment that obstruct them from remaining in a restricted set of preferred states of the world. Under this assumption, biological agents develop a variety of skills, such as perception, action, planning, and learning, that agents continuously adapt along their lives.

Active inference and the free energy principle have been employed to explain and simulate several complex processes across different disciplines. In psychology, they have been used to ground a computational account of neuropsychological syndromes [3] and to develop emotional recognition devices, which allow resolving uncertainty about emotional states by interaction and learning [4]. In economics, the free energy principle has been exploited to reformulate the agents’ optimization process in terms of their beliefs [5]. Variational approaches have been employed to explain niche construction, based on the free energy principle [6,7]. Active inference has been used to model smooth and saccadic eye movements [8,9] and to conceptualize attention [10], salience, and memory [11]. In the context of scene construction, active inference provides an explanation of how agents infer a higher-order visual pattern of a scene through visual foraging [12,13].

Under some sets of assumptions [14,15], the free energy principle can also be used to explain how all biological organisms and processes that subserve perception and action naturally emerge and are continuously adjusted through a natural model selection process. On shorter (somatic) timescales, minimizing free energy leads to the development of the single organisms’ brains, giving rise to learning and memory functions. On longer (evolutionary) time scales, free energy minimization fosters the evolution process of the species [16,17]. Establishing a statistical boundary for the Earth’s climate system, this can also be interpreted as a self-producing system that is performing active inference [18]. Systems that are capable of self-production, i.e., continuously generating and maintaining themselves by creating their own parts, are called autopoietic [19]. It can be shown that the process of autopoiesis minimizes free energy, given that for an organism to maintain a model of itself and its environment it must minimally self-produce the components required to carry out the process of preserving its generative model [20].

In active inference, the agent minimizes a variational free energy objective with respect to past experience that causes it to learn a generative model of the world, which allows predicting what will happen in the future in order to avoid surprising states. Variational inference [21], from which originates the variational free energy functional, makes possible to cast the process of perception as an optimization process, which subsumes a set of choices in terms of modeling and learning, such as the choice of the state distribution or the way the model copes with uncertainty. The generative world model is then used in the future to plan actions that will maintain the agent in a homeostatic state that satisfies the agent preferences. The agent operates an (amortized) Bayesian selection of actions that will have the least surprise with respect to its preferred state. This decision-making process takes into consideration several aspects of learning, such as epistemics, habit learning, and preference learning.

Recent developments in deep learning have opened new frontiers for studying and experimenting with different perception and behavioral theories; enabling the practical analyses of artificial implementations, either in simulations or in real environments. One popular example in this regard is reinforcement learning (RL) [22], a theory that links the dopamine signals in the brain to reward signals that can be used to reinforce correct behaviors [23,24], and describes how intelligent behaviors can be learned through reward maximization [25]. Combining RL with deep learning models for function estimation [26] has led to several empirical successes, allowing the training of artificial agents to play video games [27,28], master board games [29], or execute robotic tasks [30]. Similarly, deep learning techniques are also starting to arise in the context of active inference [31,32,33].

This work aims to survey the current state-of-the-art of deep learning models for active inference. At the same time, we want to provide a reference and a starting point for machine learning practitioners to become acquainted with active inference, drawing parallels between active inference and recent advances in RL. Other reviews of active inference have previously been presented for expressing generative models in continuous spaces [34] or discrete spaces [35]; however, the context, the practices presented, and the scope of our work strongly differ from theirs, in that we focus specifically on deep-learning-based techniques to scale active inference to large continuous state and action spaces or high-dimensional settings, such as robotics and visual control. Concurrently to our work, a review on active inference for robotics has been released [36], including references to deep learning methods for active inference. Our work differs from theirs in that they specifically focus on methods that enable applying active inference in robotics, while we more broadly discuss active inference techniques that can be employed to develop active inference artificial agents, explaining in details how each component could be implemented, highlighting the challenges to overcome, and establishing connections between methods in different machine learning areas.

A non-exhaustive summary of the aspects that are considered in the active inference framework for learning perception and action is presented in Figure 1. We divide into model learning on the one hand, and action selection on the other hand. The first comprises minimizing variational free energy for learning generative models based on past experiences, modeling belief states and priors, learning state representations and uncertainty, whereas the latter deals with selecting actions in the future, which trade-off epistemic foraging and realizing preferences, either by planning or learning habits. Two recurrent patterns in active inference are variational inference and the amortization of Bayesian selection. Variational inference allows casting inference as an optimization problem, i.e., finding the distribution closest to the actual one. Amortization allows a faster computation of the inference process, by reusing previous computations [37]. In active inference, amortized inference is applied for the choice of variational parameters, in the model learning process, and for the formation of habits, in the action selection process. Combining the two techniques strongly reduces the computational requirements of active inference and makes the framework promising for implementation in silico; however, without adequate models, it is unfeasible to scale active inference in complex scenarios, with continuous and/or high-dimensional state/action spaces.

In deep learning, generative models have been widely studied, obtaining outstanding results in several domains, such as image generation [38,39,40], text prediction [41,42,43], and video modeling [44,45,46,47]. In particular, temporal deep generative models that allow predicting the dynamics of a system, i.e., the environment or world, have been studied for control [48,49,50], curiosity and exploration [51,52,53], and anomaly detection [54]. Several of these models have been used in settings that are similar to the active inference one, and some of them even share some similarities with the active inference objective of minimizing variational free energy. As for action selection, several works that use deep learning have improved upon more classical methods (e.g., α−β pruning, A*, beam search) allowing to search much larger state and action spaces. Some examples are dynamic programming-related techniques in RL [26], evolutionary strategies [55], and Monte Carlo Tree Search (MCTS) [29,56]. These methods all combine more general and classical planning strategies with deep learning for function estimation, enabling the scaling of behavior learning and action selection for complex environments.

The remainder of this work is organized as follows: in Section 2, we present the free energy principle and explain what minimizing free energy entails in terms of perception and action; in Section 3, we establish the connection between model learning, according to active inference, and deep generative models, analyzing the different ingredients involved. In Section 4, we discuss the implementation and design choices that underlie the action selection process, relating existing works on habit learning, exploration and model-based control. Finally, we conclude our discussion, remarking the work conducted until this moment, addressing the implications of learning with deep learning, and proposing some future perspectives.

## 2. The Free Energy Principle and Active Inference

The free energy principle is at the core of the active inference framework, as it conceptualizes the development of embodied perception as the result of minimizing a free energy objective. As we show in this section, the free energy is a function of the agent’s beliefs about the environment, representing a (variational) upper bound on surprisal from sensorial stimuli. This entails that reducing free energy additionally reduces the agent’s model surprise, restricting its existence to a limited set of craved beliefs. The free energy principle originated from the work of von Helmholtz on ‘unconscious inference’ [57], postulating that humans inevitably perform inference in order to perform perception. This implies that the human perceptual system continuously adjusts beliefs about the hidden states of the world in an unconscious way. The variational formulation of the free energy [1,58], along with the introduction of actions as part of the inference process, expanded the original free energy principle leading to the development of active inference.

In Figure 2, we illustrate the interplay between the main factors that determine the embodied perception process as described in active inference. At any time, the environment is in a certain state η, which is external to the agent and not directly observable. The agent interacts with the environment in two ways: either through (passive) sensorial perception, which is characterized by the observation of sensorial states *o*, or by actions, which can be cast as a set of active states *a* that the agent imposes on the environment. According to the free energy principle, in order to minimize free energy, the agent learns an internal model of potential states of the environment. Crucially, these internal states do not need to be isomorphic to the external ones, as their purpose is explaining sensorial states in accordance with active states, rather than replicating the exact dynamics of the environment Isomorphism, in this context, refers to considering a structure-preserving mapping of the state space. According to active inference, internal and environment states are not necessarily equal and the way the internal states are organized may even differ from agent to agent, despite having to deal with similar concepts/observations/sensory states. From a biological perspective, this finds evidence in the fact that different living systems have developed different organs/tissues along their evolutionary process [16]. The role of the internal state representation is, in fact, to provide the sufficient statistics that allow a ‘best guess’ about the causes of the agent’s observations and the selection of adaptive action policies [59].

As a consequence of minimizing free energy, the agent possesses beliefs about the process generating outcomes, but also about action policies that lead to generating those outcomes [60]. This corresponds to a probabilistic model of how sensations are caused and how states should be actively sampled to drive the environment’s generation of sensory data. Because of these assumptions, the concept of ‘reward’ in active inference is very different from rewards in RL, as rewards are not signals used to attract trajectories, but rather sensory states that the agents aims to frequently visit in order to minimize its free energy [61]. From an engineering perceptive, this difference is reflected in the fact that rewards in RL are part of the environment and, thus, each environment should provide its unique reward signal, while in active inference ‘rewards’ are intrinsic to the agent, which would pursue its preferences in any environment, developing a set of most frequently visited states.

In the remainder of this section, we discuss how the agent’s probabilistic model for perception and action is learned by minimizing free energy, providing a mathematical synthesis. We consider the environment as a partially observable Markov decision process (POMDP), represented in Figure 3. Using subscripts to specify the discrete time steps, we indicate the observation or outcome at time *t* with $o_{t}$. To indicate sequences that span over an undefined number of time steps, we use the superscript ∼, i.e., for outcomes $\tilde{o}=\left\{o_{1},o_{2},\ldots ,o_{t}\right\}$. The succession of states $\tilde{s}=\left\{s_{1},s_{2},\ldots ,s_{t}\right\}$ is influenced by sequences of actions, or policies, that we indicate with $\pi =\tilde{a}=\left\{a_{1},a_{2},\ldots ,a_{t}\right\}$. Parameterization of the state-outcome likelihood mapping is indicated with θ. A precision parameter ζ influences action selection working as an inverse temperature over policies.

The section is divided in two parts: the first explains how the internal model is learned with respect to past experience, minimizing a variational free energy functional that explains the dynamics of the environment’s outcomes, given a sequence of actions. In the second part, we discuss the minimization of expected free energy with respect to the future, when actions are selected to reduce surprise with respect to the agent’s preferred outcomes. Importantly, our treatment refers to a discrete-time instantiation of active inference. For a discussion on continuous time, the reader may refer to [34].

### 2.1. Variational Free Energy

In order to minimize the negative log evidence (also known as “surprisal” or “surprise”) of observations from the external world, the agent exploits its past sensory experiences to learn a generative model of the environment using variational inference. Under the free energy principle [62,63], an upper bound on surprise is established as:(1)$$\begin{array}{cc} \underset{\mathrm{surprise}}{\underset{︸}{-\log p(\tilde{o})}} & =-\log E_{q(\tilde{s},\pi ,\theta ,\zeta )}\left[\frac{p(\tilde{o},\tilde{s},\pi ,\theta ,\zeta )}{q(\tilde{s},\pi ,\theta ,\zeta )}\right]\leq \underset{\mathrm{variational}\;\mathrm{free}\;\mathrm{energy}\;F}{\underset{︸}{E_{q(\tilde{s},\pi ,\theta ,\zeta )}\left[-\log\frac{p(\tilde{o},\tilde{s},\pi ,\theta ,\zeta )}{q(\tilde{s},\pi ,\theta ,\zeta )}\right]}} \end{array}$$
where, left to right, the following operations are performed: (i) the surprise over the sequence of observations is marginalized (i.e., summed over/integrated) with respect to the other factors of the generative model, which are the sequence of states, the policy of actions, the model parameters, and the policy precision parameter, (ii) a variational posterior over states and policies $q(\tilde{s},\pi ,\theta ,\zeta )$ is introduced for variational inference [21], (iii) Jensen’s inequality is applied to push the logarithm inside the expectation operator.

The variational free energy F can be reformulated as:(2)$$\begin{array}{cc} \; & F=E_{q(\pi ,\theta )}\left[F_{\pi ,\theta }\right]+D_{\mathrm{KL}}\left[q\left(\pi ,\zeta \right)∥p\left(\pi ,\zeta \right)\right]+D_{\mathrm{KL}}\left[q\left(\theta \right)∥p\left(\theta \right)\right],\\ \; & F_{\pi ,\theta }=E_{q(\tilde{s}|\pi ,\theta )}\left[\log q\left(\tilde{s}|\pi ,\theta \right)-\log p\left(\tilde{o},\tilde{s}|\pi ,\theta \right)\right]. \end{array}$$

When minimizing the variational free energy with respect to the past, the second equation is generally adopted as the two Kullback–Leibler (KL) divergence terms in the first equation can be neglected. The former KL divergence, referring to the policies and the precision parameter, can be overlooked because policies in the past are observed. The latter KL divergence, referring to the model parameters, can be later considered to work on top of the model through regularization techniques, e.g., similarly to sleep, where redundant synaptic parameters are eliminated to minimize model complexity [64].

Omitting conditioning on π and θ for brevity, allows the expression of the free energy in its two typical forms:(3)$$\begin{array}{cc} F_{\pi ,\theta } & =\underset{\mathrm{complexity}}{\underset{︸}{D_{\mathrm{KL}}[q\left(\tilde{s}\right)∥p\left(\tilde{s}\right)]}}-\underset{\mathrm{accuracy}}{\underset{︸}{E_{q(\tilde{s})}\left[\log p\left(\tilde{o}|\tilde{s}\right)\right]}}\\ \; & =\underset{\mathrm{approx vs true posterior}}{\underset{︸}{D_{\mathrm{KL}}[q\left(\tilde{s}\right)∥p\left(\tilde{s}|\tilde{o}\right)]}}-\underset{\mathrm{log evidence}}{\underset{︸}{\log p(\tilde{o})}} \end{array}$$

On the one hand, minimizing free energy implies maximizing the accuracy of a likelihood model $p(\tilde{o}|\tilde{s})$ while reducing the complexity of the posterior distribution. On the other hand, it implies optimizing the variational evidence bound, reminding that the KL divergence is always non-negative, namely $D_{\mathrm{KL}}[\cdot ∥\cdot ]\geq 0$ for any distribution. The KL divergence is zero when the agent’s model perfectly matches the environment dynamics, corresponding to the optimal scenario.

Though there is no expectation operator over past experiences (both for brevity and to comply with the typical way of expressing this functional), it should be clear that the agent minimizes variational free with respect to known sequences of past observations and policies from the environment. As we discuss in the following paragraphs, this is fundamental and is the main aspect that differentiates the variational free energy, computed with respect to past states of the environment, from the expected free energy, which considers future states and unobserved data.

### 2.2. Expected Free Energy

In order to minimize free energy in the future, the agent should adapt its behavior, i.e. the active states, to confine its existence within a limited set of states. These states correspond to the so called *preferences* of the agent, or preferred observations/outcomes/states, depending on the context. The objective of the agent is to exploit its knowledge about the environment, available through the internal model, to perpetually fulfill preferred perceptions in the upcoming future. While minimizing free energy about future sequences, the agent imagines how the future would look like, given a certain sequence of actions or policy. This is reflected by an expectation over future states and observations generated by the model, which exploits both the internal states model and the likelihood mapping from the environment as $\tilde{q}=q\left(\tilde{o},\tilde{s},\theta |\pi \right)=p\left(\tilde{o}|\tilde{s},\theta \right)q\left(\tilde{s}|\pi \right)q\left(\theta \right)$.

Starting from this assumption, the expected free energy G can be expressed as:(4)$$\begin{array}{cc} G_{\pi } & =E_{\tilde{q}}\left[\log q\left(\tilde{s},\theta |\pi \right)-\log p\left(\tilde{o},\theta ,\tilde{s}|\pi \right)\right]\\ \; & =-\underset{\mathrm{information}\;\mathrm{gain}\;(\mathrm{hidden}\;\mathrm{states})}{\underset{︸}{E_{\tilde{q}}\left[\log p\left(\tilde{s}|\tilde{o},\pi \right)-\log q\left(\tilde{s}|\pi \right)\right]}}-\underset{\mathrm{information}\;\mathrm{gain}\;(\mathrm{parameters})}{\underset{︸}{E_{\tilde{q}}\left[\log p\left(\theta |\tilde{s},\tilde{o},\pi \right)-\log q\left(\theta \right)\right]}}-\underset{\mathrm{extrinsic}\;\mathrm{value}}{\underset{︸}{E_{\tilde{q}}\left[\log p\left(\tilde{o}\right)\right]}} \end{array}$$

Hence, minimizing the expected free energy implies that the agent: (i) maximizes epistemic value, i.e., mutual information between hidden states and sensory data, (ii) maximizes parameter information gain, i.e., mutual information between parameters and states, and (iii) maximizes extrinsic value, i.e., the log likelihood of outcomes under a preferred, prior distribution $\log p(\tilde{o})$, or rewards.

Again, omitting conditioning on the policy for brevity, it is possible to rewrite the expected free energy as a minimization of risk and ambiguity: (5)$$G_{\pi }=\underset{\mathrm{ambiguity}}{\underset{︸}{E_{\tilde{q}}[H\left[p\left(\tilde{o}|\tilde{s}\right)\right]}}+\underset{\mathrm{state}\;\mathrm{control}}{\underset{︸}{D_{\mathrm{KL}}[q\left(\tilde{s}\right)∥p\left(\tilde{s}\right)]}}-\underset{\mathrm{information}\;\mathrm{gain}\;(\mathrm{parameters})}{\underset{︸}{E_{\tilde{q}}\left[\log p\left(\theta |\tilde{s},\tilde{o}\right)-\log q\left(\theta \right)\right]}}$$
(6)$$\begin{array}{cc} \; & \approx \underset{\mathrm{ambiguity}}{\underset{︸}{E_{\tilde{q}}[H\left[p\left(\tilde{o}|\tilde{s}\right)\right]}}+\underset{\mathrm{risk}}{\underset{︸}{D_{\mathrm{KL}}[q\left(\tilde{o}\right)∥p\left(\tilde{o}\right)]}}-\underset{\mathrm{information}\;\mathrm{gain}\;(\mathrm{parameters})}{\underset{︸}{E_{\tilde{q}}\left[\log p\left(\theta |\tilde{s},\tilde{o}\right)-\log q\left(\theta \right)\right]}} \end{array}$$

Here, we assume the bound is tight and the approximate posterior is a good approximation of the true posterior to express risk in terms of outcomes; however, the agent may also express its homeostatic preferences in terms of internal states rather than on its sensorial perceptions, and formulate the expected free energy in terms of state control.

Despite the variety of objectives that can be considered for the expected free energy, by either changing some assumptions or reordering its factors, the imperative of the functional stays the same. The goal of the agent is to restrict itself to its preferred set of states/sensorial perceptions, while minimizing the ambiguity of its internal model. Maximizing epistemic value and/or parameter information gain, as for Equation (Equation 4), also implicitly adheres to this hypothesis, as finding informative states in the environment will minimize the uncertainty of the model in the future [63,65].

## 3. Variational World Models

In active inference, the objective of minimizing surprise of the internal model with respect to sensory inputs induces a continuous model learning process that happens inside the brain. This assumes a predictive coding interpretation of the brain, where the internal model is used to generate predictions of sensory inputs that are compared to the actual sensory inputs. The internal model attempts to explain the dynamics of the world and thus, as performed in related work [50,66], we also refer to it as the ‘world model’. The reaction of the internal model, tending to minimize free energy with respect to the sensory inputs, accounts for perception of the agent, which learns to predict the sensory inputs and the causal structure of their generation.

In machine learning, such a learning process, which requires no human supervision or labelling, is generally referred to as self-supervised or unsupervised learning. In contrast to biological agents, deep learning systems typically use a batch learning scheme, where a dataset of past trajectories is collected, and models are parameterized as deep neural networks that are optimized by training on batches of data sampled from this dataset. Such a dataset, indicated by $D_{\mathrm{env}}$, can be seen as an ordered set of triplets containing an environment observation, the agent’s action, and the following observation (caused by the action), namely the triplet $\left(o_{t},a_{t},o_{t+1}\right)$. Training the model is then typically alternated with collecting new data by interacting with the environment, using the model for planning [49], or using an amortized (habitual) policy [33,67]. In practice, one can also train a model upfront using a dataset of collected trajectories from a random agent [68] or an expert [32]. The latter is especially relevant in contexts where collecting experience online with the agent can be expensive or unsafe [69].

The free energy loss to minimize for one time step, i.e., $\pi =a_{t}$, can then be written under the expectations of the data (observations and actions) from the replay buffer:(7)$$F_{a_{t}}=E_{\;D_{\mathrm{env}}}\left[D_{\mathrm{KL}}\left[q\left(s_{t+1}|s_{t},a_{t}\right)∥p\left(s_{t+1}|s_{t},a_{t}\right)\right]-E_{q(s_{t+1}|s_{t},a_{t})}\left[\log p\left(o_{t+1}|s_{t+1}\right)\right]\right]$$

Given the above loss, three distributions or models need to be instantiated: (i) a likelihood model $p(o_{t+1}|s_{t+1})$ that allows generating (also known as reconstructing) sensory data from the model’s internal states, (ii) a prior model $p(s_{t+1}|s_{t},a_{t})$, which encodes information about the transition probabilities of the dynamics of the internal states, (iii) a posterior model $q(s_{t+1}|s_{t},a_{t})$, which is chosen to minimize the upper bound on surprise, according to variational inference [21]. In machine learning, this formulation is better known as the (negative) evidence lower bound (ELBO), it is the same as the loss used to train variational autoencoders (VAE) [70,71]) and it is shown to optimize a variational information bottleneck, trading off between the accuracy and the complexity of summarizing the sensory information in the internal state representation [72,73]. As it is assumed that external states are not observed by the agent and that the dynamics of the environment is unknown, there is no way to ensure that the representation learned is the same as external one; however, the relationship with information compression techniques ensures that minimizing free energy entails optimizing the information contained in the internal states about the sensory states [62,74].

Theoretically, deep learning models can be employed to approximate any function with an arbitrary degree of accuracy [75], which in our case means predicting both the internal states and sensory data distributions with an arbitrary degree of accuracy. In practice, though, obtaining an highly accurate model is difficult and, while neural models can find useful approximations of these models, if one of them is well-known in advance, directly using it and adapting the other models accordingly could lead to more satisfactory results. For instance, if the actual likelihood model is known for a certain state space, the state space of the prior and posterior models can be adapted accordingly. This is the case for differentiable simulators, where the environment’s observations are the result of a differentiable generation process that can be integrated in the world model [76,77]. Or again, if the dynamics of the environment is known, it is possible to use that as a prior, forcing the internal states of the model and the external states of the environment to have the same structure. When the environment is represented as a POMDP, having complete knowledge of the dynamics is the only case that ensures an optimal behavior can be found [22,78]; however, even knowing the dynamics in advance, solving the POMDP problem remains computationally intractable. Function approximations techniques and procedures to construct an improved state representation, as the model learning approaches we describe here, are often used to find nearly optimal policies in more computationally efficient ways [79,80,81].

In a general scenario, all distributions are unknown and must be either learned by the agent or assumed having a certain form, according to some design choices. There are indeed numerous options to consider when instantiating the different models, and some of them are important to carry out a stable optimization and/or a well-thought amortization of the Bayesian inference process. Other design choices consider different aspects of the generative model (Figure 3), such as the parameters of the likelihood mapping θ and the sensitivity of the model, which are certainly relevant but that can often be assumed fixed and neglected.

### 3.1. Models

To instantiate the deep neural networks for the agent’s generative model, first, it is important to consider the nature of the variables involved. For the hidden states, active inference assumes a probabilistic model. Unless the environment state space nature is known, the internal state distribution may have no predefined structure, and neural networks can be trained to output distributions of different kinds; however, in order to compute the expectations, as in Equation (Equation 7), it is important that the sampling process of the distribution is differentiable, as the objective needs to be backpropagated through the model for computing the gradients that update the model [82]. As the sampling process is generally non-differentiable, the gradients of the samples should be estimated with ad hoc techniques. Some widely known examples are the reparametrization trick for Gaussian distributions [70,71], the straight-through gradient method [83], the likelihood-ratios method [84], also known as REINFORCE gradients [85], for Bernoulli and categorical variables.

In the active inference literature, multivariate Gaussian (also known as normal) distributions with a diagonal covariance matrix have been largely adopted since the initial works on VAEs [32,33,68,86]. Similarly, numerous latent state space models have adopted a Gaussian structure of their latent space [48,49,87,88], but also more complex mixture models have been proposed [50]. For Bernoulli and categorical distributions, there has been both work on general-purpose generative models, such as discrete VAE [38,89], on latent dynamics models for planning [66,90], and recently they have also been used in an active inference setting [91]. Some other alternatives to the above methods, which have yet to be explored for training world models, are: piecewise distributions [92], Markov chains [93], and normalizing flows [94].

**Posterior Model.** The choice of the distribution is particularly important for the posterior model, which is the variational distribution. In theory, one could search for an optimal distribution of parameters for each of the environment’s transitions/observations, though that is a slow and difficult process. To speed up training, but also guaranteeing a legitimate choice of the posterior, it is possible to amortize the selection of the posterior parameters, as presented in the original VAE work [70,71]. The autoencoding amortization scheme employs the observation corresponding to a certain state to infer the parameters of the variational distribution, $q(s_{t+1}|s_{t},a_{t},o_{t+1})$. This allows optimizing the parameters of the posterior to compress information optimally, as the posterior has access to the observation that the likelihood model wants to generate. In VAE terms the posterior model is typically called the “encoder”, whereas the likelihood model is dubbed the “decoder”.

The choice of the encoder architecture, which allows the flow of information from observations to the posterior, depends on the environment. For instance, for two-dimensional matrices of data, such as images, convolutional neural networks (CNN) [95], or other architectures for computer vision such as vision transformers [96] are common choices. Other potentially useful models can be multilayer perceptrons (MLP) for vector-structured data [97] or graph neural networks, for graph-structured data [98]. Similarly, the choice of the likelihood model depends on the format of the observation data, e.g., a transposed CNN can be useful in the case of visual data.

**Prior Model.** The prior model can either be known or learned. To learn the prior model, one can adopt a recurrent neural network architecture, i.e., using memory cells such as long short term memories (LSTM) [99] or gated recurrent units (GRU) [100]. In other cases, the environment dynamics is known upfront, or assumptions about the prior can be made, such as assuming that the prior is a uniform probability distribution. For instance, an isotropic multivariate Gaussian N(0,I), with zero mean and an identity covariance matrix *I*, can be employed as a fixed prior, as performed in standard VAE architectures [70,71]. Alternatively, assuming the laws that govern the dynamics are known (e.g., physics laws), the environment’s physics could be exploited as a strong prior [101]. In a similar fashion, in [102], the authors used the internal state of the robot to force a known prior structure on the posterior. Finally, the prior could also be ignored/considered constant, treating the model as an entropy-regularized autoencoder [103].

### 3.2. Uncertainty

In the active inference perception model, precision, or sensitivity [104,105], is generally associated to the uncertainty of the transitions between hidden states of the prior (beliefs precision) or the mapping from hidden states to outcomes of the likelihood (sensory precision), and can be expressed as the inverse variance of the distribution [106]. In active inference implementations, precision has been employed as a form of attention, to decide on which transitions the model should focus on learning from [33], though this aspect has been generally less studied in the literature. Similar mechanisms of precision have been employed for VAE models to control disentanglement of the latent state space [107] or posterior collapse [108].

Another source of uncertainty in the model arises from uncertainty in the parameters. In the deep learning community, uncertainty about model parameters has been studied employing Bayesian neural networks [109], dropout [110], or ensembles [111], and used in RL to study the exploration problem [53,112,113]. In active inference, considering the generative model in Figure 3, uncertainty is treated with respect to the distribution over the parameters of the likelihood model. Similarly to RL, dropout [33] and ensembles [114] have been studied in the active inference literature, although several implementations until now have neglected this aspect, assuming confidence over a single set of parameters.

### 3.3. Representation

Following the variational free energy formulation from Equation (Equation 7), the agent’s generative model is assumed capable of generating imaginary outcomes that match closely with the sensory perceptions through the likelihood model. This is presented on the left in Figure 4, for an environment with visual sensory data, i.e., images. The model depicted uses a sequential VAE-like setup, with the posterior encoding information from the observation (red) in the state, and the likelihood model generating observations from the state with a decoder (blue).

However, learning a likelihood model from high-dimensional sensory observations, such as in pixel-based environments, is not a trivial problem. In this case, the likelihood model needs to generate images that match with the original observations pixel by pixel, requiring both a high-capacity model and a considerable accuracy, especially for high-resolution images. Most often, the probability distribution of an image is represented as a product of independent Gaussians over each pixel with fixed standard deviation, in which the log-likelihood loss in Equation (Equation 7) becomes the pixel-wise mean squared error between two images; however, this can be problematic, as it might lead to the model ignoring small but important features in the environment (as the pixel-wise loss is low) and wasting a lot of capacity in encoding potentially irrelevant information (i.e., the exact textures of a wall).

One potential solution (a) is to train the model in sight of the future task to accomplish, considering only the states related to accomplishing the agent’s goal, i.e., rewards. Finding a hidden state representation that allows predicting such information, without necessarily generating observations greatly lightens the representational burden of the model, though it would make the internal dynamics less informed. An example is illustrated in Figure 4a. Similar representations have been employed for RL [29], and could perhaps be also adapted for active inference.

Another proposed solution (b) is to replace the likelihood component of the loss with a state-consistency loss. These kinds of representations, which enforce some form of consistency between states and their corresponding sensory observations, have increasingly gained popularity in deep learning as self-supervised learning methods, such as contrastive learning [115], clustering/prototypical methods [116], distillation/self-consistency methods [117,118], and redundancy reduction/concept whitening [119,120]. These representation techniques, depicted in Figure 4b, have also been shown successful in training dynamics models for RL [121,122] and, recently, for active inference as well [123].

Rather than replacing the likelihood model, some other approaches (c–d) have instead focused on improving the capacity of the model. This is the case of the memory-equipped models (Figure 4c) and hierarchical models (Figure 4d). Using memory allows to preserve more information about other (past) observations, and has shown encouraging results in training latent dynamics models with deep learning memory models, such as LSTMs and GRUs [47,48,49,124]. The memory increases the capacity of the model and allows more accurate predictions of states that are far in the future, especially when the prior model is unknown and must be learned.

Hierarchical models, which in the active inference community are also referred to as deep active inference models [13,125] (with an unfortunate confusion caused by using “deep active inference” as a term for active inference methods using deep neural networks in the generative model [31,32,102,126]), use a multi-layer structure of the hidden states of the model that facilitates the modeling of part-whole or temporal hierarchies. Similarly to using a memory, a hierarchy of states can increase the representational capacity of the model and allow more accurate predictions. Some deep learning examples have already implemented this [127,128] as well as some active inference implementations for long-term navigation [69].

### 3.4. Summary

There are several choices to consider when designing a variational world model. In this section, we explain some of them along with providing references to a variety of studies that actually implement these mechanism, considering contributions both in the larger deep learning literature as well as in an active inference context. A summary of the design choices is presented in Table 1.

In the future, it will be important to continue investigating these design choices and analyzing their interplay. We believe that synergies between advances in the deep learning community and active inference adopters will be crucial to further develop generative models for a wide range of use cases. We also look forward towards novel aspects of model learning, such as the representation of time [133] or the model reduction happening during sleep [134].

It is also important to note that some of the design choices discussed, such as the internal state representation, have an impact on the model learning part but also on the agent’s action selection. For instance, if there is no likelihood model, how should the agent recognize whether the preferred outcomes are being satisfied? Or, again, if the agent’s model is hierarchical, how should granular actions relate to the hierarchical state structure? In the next section, we provide an overview of the techniques proposed to adopt the generative model for action selection, and elaborate on these issues.

## 4. Bayesian Action Selection

In order to select future actions in active inference, the agent exploits the learned model in order to match its preferred outcomes by minimizing the expected free energy. More formally, the agent’s belief over which policy or sequence of actions to follow is given by:(8)$$p\left(\pi |\zeta \right)=\sigma \left(-\zeta G_{\pi }\right),$$
where σ is the softmax function and ζ a precision parameter. Hence, when precision is high, the agent is most likely to engage in the policy with the lowest expected free energy, whereas for (very) low precision the agent will rather randomly explore. As discussed in Section 2, the expected free energy $G_{\pi }$ is calculated by taking expectations with respect to outcomes in the future, inside the model’s predictions. Hence, the agent minimizes the expected free energy by evaluating the predicted outcomes against the preferred distribution before deploying actions in reality.

Whereas the generative model is trained to match the real outcomes of the world with past experience, future outcomes are not yet available to the agent. As an active inference agent adopts prior expectations of reaching preferred outcomes, one can interpret this as having a biased generative model of the future towards one’s preferences. The self-evidencing behavior that emerges is that of a ‘crooked scientist’ [7], searching active states that will provide evidence for its biased hypothesis.

From a biological perspective, we could assume that every agent possesses a unique set of preferences, i.e., to maintain homeostasis [135]. These preferences could, for instance, associate internal signals, such as body temperature, hunger, happiness, and satisfaction, to the preferred states of the world. For artificial agents, defining the correct set of preferences can instead be problematic. Different ways of addressing this problem are presented in the first subsection. We also analyze the problem of dealing with the agent’s uncertainty and how to learn and/or amortize the action selection process.

### 4.1. Preferences Modeling

As summarized in Section 2, the expected free energy objective can be factored in several ways, each highlighting different emergent properties of the agent’s behavior (Equations (Equation 4)–(Equation 6)). While this aspect of active inference has been the target of critics [136], this allows for greater flexibility in designing the agent selection process.

**Observation Preferences.** If the agent’s objective is to match a set of preferred outcomes, the preferred distribution is over the environment’s observations p(o). Matching outcomes can be seen as a form of goal-directed behavior, where the agent plans its actions to achieve certain outcomes from the environment. Goal-directed behavior has been widely studied in the context of RL, both in low-dimensional [137] and visual domains [138,139]. Preferences defined in the observation space can be handy, as they just require observations from “snapshots” of the environment in the correct state. Nevertheless, artificial active inference implementations have rarely used them, as they are generally hard to match in the high-dimensional settings. Strategies that overcome such limitations [123] could be the subject of future studies.

**Internal State Preferences.** Instead of defining preferences in observation space, these could be directly instantiated in the internal state space of the agent. This form of state matching [140] assumes that the agent knows both the preferred states distribution p(s) and the model in advance, or as typical in RL, that sensory states are used as internal states. Alternatively, if a set of preferred outcomes is available, preferred states can be inferred from those using an inference model p(s|o). This approach has been applied in robotics simulated and realistic setups [32,68].

**Rewards as Preferences.** Another way to circumvent the problem of defining preferences is to use a reward function that represents the agent’s probability of observing the preferred outcomes. The RL problem can be cast as probabilistic inference, by introducing an optimality variable $O_{t}$, which denotes whether the time step *t* is optimal [141]. The distribution over the optimality variable is defined in terms of rewards as $p\left(O_{t}=1|s_{t},a_{t}\right)=\exp\left(r\left(s_{t},a_{t}\right)\right)$. As discussed in [142], RL works alike active inference but it encodes utility value in the optimality likelihood rather than in a prior over observations. Assuming $\log p\left(o_{t}\right)=\log p\left(O_{t}|s_{t},a_{t}\right)$, the environment rewards can be used for active inference as well. This possibility has allowed some active inference work [33,114] to reuse reward functions from RL environments [22]. Concretely, it is possible to consider rewards as a part of the observable aspects of the environment, and define their maximum values as the preferred observations [143]. Nonetheless, defining reward functions is also problematic [144] as they are not naturally available, and this setup works well only for well-engineered environments.

**Learned Preferences.** Finally, state preferences can also be learned from previous experience using conjugate priors [91], or from expert demonstrations [68]. In a RL context, demonstrations can be used in an inverse reinforcement learning fashion [145,146], where a reward signal is inferred from correct behaviors, which is then optimized using RL techniques.

### 4.2. Epistemics, Exploration, and Ambiguity

While active inference agents seek to realize their preferences, they also aim to reduce the uncertainty of their model. For instance, if an agent has to manipulate some objects in a dark room, it would first search for the light switch to increase the confidence of its model and reduce the resulting ambiguity of its actions. As also shown in Section 2, the causes of the agent’s ambiguity can be twofold: on the one hand, it can be due to the incapability of inferring its state with certainty, referring to uncertainty in the state-observation mapping, e.g., likelihood entropy or mutual information; on the other hand, the uncertainty can be caused by the agent’s lack of confidence with respect to the model’s parameters. As depicted in Equation (Equation 4), an agent’s drive for epistemic foraging is caused by maximizing two information gain terms: information gain on model parameters and information gain on hidden states.

**Parameter-driven Exploration.** Maximizing mutual information in parameter space has been studied in RL as a way to encourage exploration, computing the information gain given by the distribution over parameters with ensembles [112,113,147] or Bayesian neural networks [53]. In particular, in [113], they use the model to both evaluate the states/actions to explore and to plan the exploratory behavior, which is close to what envisioned in active inference. Ensemble methods have also been employed in some active inference works [114,129] along with dropout [33].

**State-driven Exploration.** Maximizing mutual information between states and observations has also been studied in RL for exploration, using the Bayesian surprise signal given by the D_KL_ divergence between the (autoencoding) posterior and the prior of the model as a reward [51]. Alternatively, the surprisal with respect to future observations has also been used in RL to generate an intrinsic motivation signal that rewards exploration [52,148,149]. In active inference, the majority of works have instead focused on using multiple samples from the likelihood model [32,33].

**Uncertainty Tradeoffs.** It is worth mentioning that, during different stages of training, uncertainty related to parameters and uncertainty related to sensory/internal states may overlap. Particularly, given that the distributions that represent the agents’ states are inferred by employing the model parameters, uncertainty in the model strongly influences uncertainty with respect to the state. This highlights the importance of considering both kinds of uncertainty, especially when the model is imperfect or its learning process is incomplete.

From an engineering perspective, having to deal with multiple signals, as in the active inference objective, poses additional optimization problems. Different parts of the objectives may provide values on different scales, depending on the different models, distributions and on the sensory data processed by the agent. In the RL landscape, how to combine the environments’ rewards with exploration bonuses to obtain the best performance is an ongoing research problem. While one might consider using the ‘vanilla’ objective, with no weighting of the different components, weighting could lead to different behaviors that might come in use for practical purposes, e.g., reducing the weight on the exploration/ambiguity terms might lead to faster convergence when there is no ambiguity/need to explore in the environment.

### 4.3. Plans, Habits, and Search Optimization

From a computational point of view, the most complex aspect of minimizing the expected free energy consists of how selecting the actions that will accomplish the agent’s belief. In practice, optimizing $G_{\pi }$ becomes a tree search, optionally pruning away from the search all policies that fall outside of an Occam’s window, which are the policies that have a very low posterior probability. Nonetheless, depending on the way policies are defined, the search can still be significantly expensive, especially in high-dimensional and continuous actions domains.

We distinguish three ways of establishing action selection, summarized in Figure 5. The first is the typical active inference’s definition, with the policy being a sequence of actions $\pi =\{a_{1},a_{2},a_{3},\ldots \}$, and we will refer to these policies as *plans* for distinction (Figure 5a). Each plan is evaluated by its expected free energy, and the next action is selected from the best plan according to Equation (Equation 8). The second way of defining a policy is by learning a state-action mapping $\pi (s_{t})$, which is amortizing policy selection by finding an optimal *habit* policy that outputs the expected best action for each state (Figure 5b). This is also the notion of a policy that is accustomed in a typical RL setting. Finally, it is possible to combine both worlds by first estimating the expected free energy for a given state and action, and then performing a search over the reduced search space (Figure 5c).

**Plan-based policies.** Assuming a complete search over all potential sequences of actions, the plan-based method should yield the optimal policy. Unfortunately, in most domains, considering all sequences of actions is an intractable problem and more engineered random shooting methods are used to search only over the most promising sequences of actions, such as [55]. Similar methods have been employed both for RL [49] and active inference [32,68]. In particular, when the search over policies takes into account recursive beliefs about the future, this scheme is referred to as sophisticated inference [74]. Sophistication describes the degree to which an agent has beliefs about beliefs. A sophisticated agent, when evaluating a sequence of actions, instead of directly considering the sequence of outcomes, recursively evaluates outcomes in terms of the beliefs it would have when applying each action of the sequence.

**Habit Policies.** For habit policies, we consider a one-action version of the expected free energy G that can be obtained by considering one-action plans $\pi =a_{t}$ for all time steps:(9)$$\begin{array}{cc} G_{a_{t}} & =E_{q(s_{t+1},o_{t+1}|s_{t},a_{t})}\left[\log q\left(s_{t+1}|s_{t},a_{t}\right)-\log p\left(o_{t+1},s_{t+1}|s_{t},a_{t}\right)\right] \end{array}$$

A state-action policy $\pi (s_{t})$ can then be trained to maximize the above signal over multiple time steps, as typically performed in policy gradient methods [150,151]. In order to plan for longer horizons, deep RL methods adopt value functions that allow estimating the expected sum of rewards over time, over a potentially infinite horizon. For these long-term estimates, the value functions utilize a dynamic programming approach, where values are continuously updated bootstrapping current estimates with actual data. From an active inference perspective, it is also possible to estimate the expected free energy for a longer horizon by applying dynamic programming, similarly to what was studied in [126]. The expected free energy can then be rewritten and optimized recursively as follows:(10)$$G_{\pi }\left(s_{t},a_{t}\right)=G\left(s_{t},a_{t}\right)+\gamma E_{a_{t+1}∼\pi \left(s_{t+1}\right)}\left[G_{\pi }\left(s_{t+1},a_{t+1}\right)\right],$$
where $G_{\pi }$ represents an estimate of the expected free energy following the policy π and the expectation over π means the actions are sampled from the state-action policy distribution. The above equation resembles the Bellman equation known from RL with gamma being an (optional) discount factor that is used to avoid infinite sum. This optimization scheme leads to an habit policy that can achieve optimal behavior, when the sources of uncertainty of the environment are stationary; however, habitual learning can be insufficient in realistic scenarios, where rare and unexpected events are common. In this action selection scheme, the precision parameter ζ controls the entropy of the state-action policy distribution, similarly to maximum-entropy control approaches [151,152].

**Hybrid Search Policies.** Finally, hybrid search schemes (c) combine the use of a learned prior with computing the expected free energy for sequences of actions. The search space is greatly limited by using the prior, which influences the choice of the nodes to select and expand. One of the most popular applications in RL of these methods is by employing variants of Monte Carlo Tree Search (MCTS) [29,153], which use both a prior over actions and estimates of the expected utility over long horizons, as in Equation (Equation 10). Similar approaches have recently been applied for active inference [33,154]. While these methods are generally applicable only for discrete action spaces, extensions of MCTS for continuous domains have been developed as well [56]. The precision parameter ζ in these methods can be used to control the influence of the prior relative to the expected free energy (computed a posteriori, with respect to a certain action/plan).

### 4.4. Summary

Similar as when designing the agent’s model, there are several aspects to consider for implementing action selection in an active inference fashion. In this section, we covered a number of important aspects and provided references to existing implementations, both in the active inference and in the deep reinforcement learning landscape. A summary of these methods is presented in Table 2.

There are still several challenges to overcome in addition to the discussion so far. For instance, defining the preferences for an artificial agent is still unresolved for many practical applications. Future work should also address hierarchical implementations for action selection, to accompany hierarchical models [133], allowing to amortize and abstract the action selection further. Another interesting interesting avenue consists of investigating episodic control (currently less studied in the active inference literature), since this has played an important role for improving performance in RL [156].

## 5. Discussion and Perspectives

Developing artificial intelligence is a complex and intriguing problem. Among the capabilities that artificial intelligent agents should possess, the ability to sense and to act consistently is crucial. Intelligent agents should be able to exhibit their intelligence, manipulating the environment according to their will or purpose, and to understand the consequences of their actions, in order to provide a closed-loop feedback to their acting system and to acknowledge the accomplishment of their desires.

Active inference is a neuro-inspired framework that encompasses both a perception process, through learning a variational world model, and a Bayesian action selection process, which considers both preference satisfaction and uncertainty in the environment and in the agent’s model. The variational inference optimization scheme and the amortization of inference, when learning the model parameters and selecting actions, make the framework promising for practical implementations, however, without scalable models, it is unfeasible to apply active inference in complex scenarios, with continuous and/or high-dimensional state/action spaces. We showed how active inference can be combined with deep learning models for function approximation to provide implementations that scale to more complex environments, with the potential of applying it in realistic scenarios.

As mentioned in Section 3, neural networks can learn to approximate any function with an arbitrary degree of accuracy. Considering multi-layer perceptron models, this should theoretically achievable by using shallow networks with two hidden layers and sufficient capacity; however, empirical evidence has shown that deep neural networks tend to converge to more accurate models more easily than shallow networks [157] and that overparameterized neural networks generally lead to more accurate predictions [158]. One of the major issues of deep learning might be attributed to its gradient-based learning rule. While there are, indeed, proofs that an arbitrarily accurate neural network might exists, there is no certainty about a way to find the accurate model given a set of data, and thus the learning problem might be undecidable [159,160]. Nonetheless, we believe that the strong empirical results obtained by using deep neural networks justify their practical utility for implementing active inference. Furthermore, several of the modeling choices here discussed, such as the definition of the internal state distribution or the modality to amortize the action selection process, might be useful in other learning/modeling frameworks as well.

One of the intentions of this work is to provide an introduction to active inference and guidelines for deep learning researchers to hit the ground running in a given field by exploiting concepts that are in common between the two areas. At the same time, this article can be used as a reference for scientists intending to address some of the issues that hinder artificial implementations of the active inference framework.

We presented several design choices that need to be addressed to instantiate artificial active inference agents with deep learning models, attempting to relate them to well-established studies in both fields. In particular, we found that some aspects of active inference are well reflected in some areas of deep learning, such as unsupervised learning, representation learning, and reinforcement learning, whose findings can be used to push the boundaries of active inference further. In turn, active inference provides a framework for perception and action, from which individual approaches could obtain insights to both expand their scope or understand the implications of their work from a larger perspective.

## Figures and Tables

**Figure 1 entropy-24-00301-f001:**
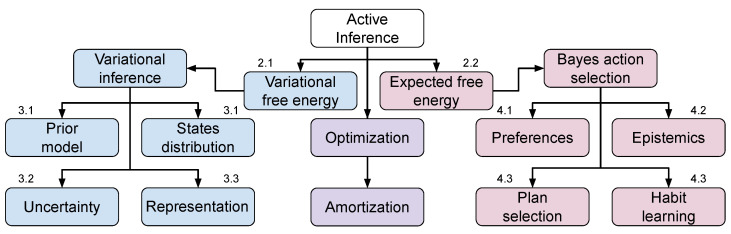
The free energy functional minimized by active inference takes two forms: variational free energy, with respect to past experience, and expected free energy, for selecting future behaviors. For each of the two, an (amortized) Bayesian optimization scheme is followed that needs to consider several aspects, as summarized in the diagram. The numbering indicates the section of the paper discussing each aspect.

**Figure 2 entropy-24-00301-f002:**
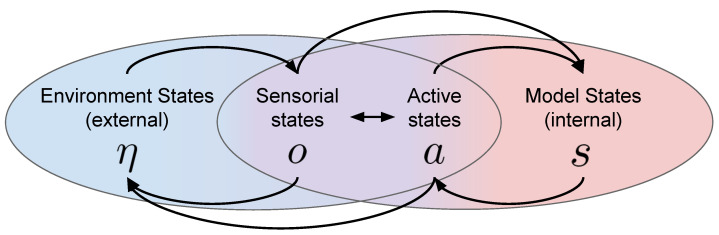
The external environment states η are the hidden causes of sensorial states *o* (observations). The environment attempts to represents such hidden causes through its internal model states *s*. Crucially, internal states may or may not correspond to external states, which means that hidden causes in the brain do not need to be represented in the same way as in the environment. Active states *a* (actions), which are developed according to internal states, allow the agent to condition the environment states.

**Figure 3 entropy-24-00301-f003:**
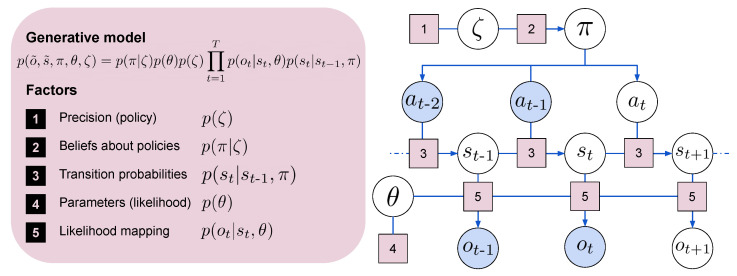
The diagram illustrates the interplay between the different factors that compose the graphical model. (1) Policy precision; (2) beliefs about policies; (3) transition probabilities, also known as dynamics; (4) parameters of the likelihood mapping; (5) likelihood model.

**Figure 4 entropy-24-00301-f004:**
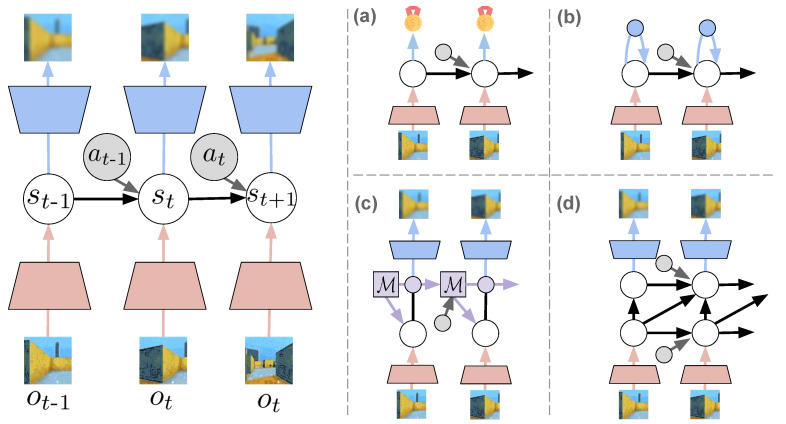
Representation learning approaches for world models with a latent dynamics. On the left, the base approach with the likelihood-model that reconstructs sensory information. On the right: (**a**) Task-oriented representation; (**b**) State-consistent representation; (**c**) Memory-equipped model (memory cell indicated with M); (**d**) Hierarchical states structure.

**Figure 5 entropy-24-00301-f005:**
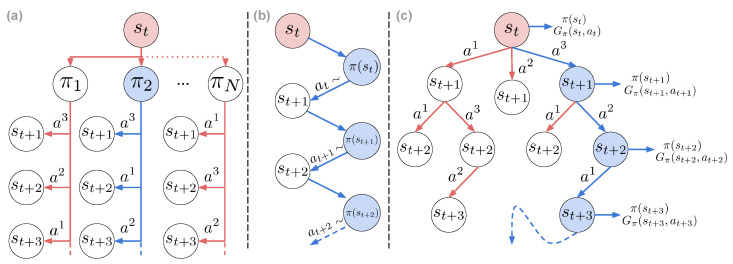
Different approaches for selecting actions. Blue circles represent the path selected by the agent. (**a**) Deep search via action plans: the path selected has the lowest free energy. (**b**) Habit learning via state-action policies: the agent always samples from the same conditional distribution. (**c**) Tree search guided by value and policy: the agent selects the actions according to the prior and the expected free energy.

**Table 1 entropy-24-00301-t001:** Implementation and design choices for learning the variational world model of the agent. The table displays one or two examples for each aspect–modality pair, both in the active inference and in the more general active inference literature, when applicable.

	Modality	Active Inference	Deep Learning
**States distribution**	Gaussian	[32,33,114,123,129]	[49,87]
	Categorical	[91]	[66,90]
	Others	-	[92,93,94]
**Prior model**	No prior	-	[103]
	Known prior	[86,102,114]	[70,71,101]
	Learned prior	[31,33,68,91,123]	[48,49,66]
**Uncertainty**	Precision	[33]	[107,108]
	Ensemble	[91,114]	[112,113]
	Dropout	[33]	[130]
**Representation**	Task-oriented	-	[29]
	State consistency	[123]	[116,119,121,122,131]
	Memory-equipped	[32,124]	[47,48,49,50,66]
	Hierarchical	[132]	[127,128]

**Table 2 entropy-24-00301-t002:** Implementation and design choices for the action selection process, minimizing the expected free energy. The table displays one or two examples for each aspect–modality pair, both in the active inference and in the deep learning (mainly, reinforcement learning) literature, when applicable. * All active inference methods generally consider hidden states exploration.

	Modality	Active Inference	Deep Learning
**Preferences**	Observations	[123]	[137,138,139]
	States	[68]	[140]
	Rewards	[33,114,126,129]	[66,67,87,155]
	Learned	[91]	[145,146]
**Exploration**	Hidden states	*	[51,52,148,149]
	Likelihood parameters	[33,91,114]	[112,113]
**Action selection**	Action plans	[32,68]	[55]
	State-action policy	[123,126]	[150,151]
	Amortized search	[33,154]	[29]

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
