# Peer review of "The Free Energy Principle for Perception and Action: A Deep Learning Perspective"

_entropy, 2022, doi:10.3390/e24020301_

Round 1

Reviewer 1 Report

The authors present a high-level literature survey of active inference (AIF), or rather, certain aspects of it, and related deep learning (DL) methods/analogues, specifically attempting to build connections between various aspects of unsupervised/representation learning and reinforcement learning. Additionally, the manuscript describes how deep RL approaches have benefited active inference methods/models and (to some degree) vice versa.

I think, overall, the survey is fine and useful and particularly liked its comments on future directions and discussion of some of the more interesting (and lesser-explored) methods/approaches that were DL-based and AIF-like, such as the "memory-equipped" VAE action-driven models.
After reading through and marking up the survey (in red pen), my thoughts are as follows.
I start with its strengths:
1) the survey is interesting and potentially useful to the deep learning and AIF communities, 
2) it usefully presents and breaks down some of the high-level equations related to AIF (such expected and variational free energy) and includes some intuitive explanations of various components of these mathematical equations (without dwelling or bloating the manuscript with derivations or dense Bayesian mechanics assuming expert knowledge/background from the reader)
and end with its weaknesses:
1) the article feels misplaced in the label/tag for Entropy under "Entropy and Biology" -- if one reads the page related to this tag, this work is quite far from this section of Entropy's goals and the subject of its articles, even though I understand its for the special issue on Active Inference. I criticize this mismatch a bit b/c the survey, while it does briefly touches on biological aspects of AIF, it does not do this part much justice (and AIF is all about the connections/analogues to human behavior, neural function, and cognitive function), at least in my view. I advise that a detailed (not necessarily very long, perhaps 1/2-3/4 of a page) section (at least) at least making some connections to the behavioral/neurobiological side of AIF (at least some of the key findings), as the original central works related to AIF and free energy minimization often do. This will help better fit the theme of this section of the journal.
Otherwise, while I recognize the value/utility of this survey, it feels a bit too strongly DL-oriented and would be published in a more technical journal in the area of machine learning, especially given that this manuscript emphasizes both at the start and end that one of its central goals is to help DL practitioners adopt/work on AIF (the flip side, AIF -> DL, would be better served by adding some nice connections to behavior/biology, perhaps including some discussion of the probabilistic, non-neural graphical models often designed in this space).
2) Some further organization/sub-division in certain sections would be useful in a survey-like context -- specifically, I would like just a bit more of a finer-grained discussion/breakdown in section 4, for example the "Epistemics, Exploration, and Ambiguity" sub-section (4.2) is just a bit too short and yet there a lot of important methods that fall under this category and same goes sub-section 4.3 (and 4.1). I recommend expanding some of the paragraphs that try to group some of the related methods with \paragraph{} or \subsubsection{} subheader (in the likely source latex) with a meaningful name to the family/group of methods a few more citations and some explanations of the approaches taken, as these portions of DL/AIF are sometimes the most important to understand/implement right for successful AIF models (and will better encourage future research in these areas).
3) The article's language needs a lot of work, while overall the flow is fine and most of the explanations are fine, looking at the red markings post-reading, this article is riddled with grammatical issues and writing flaws, which distract the reader somewhat as parsing some sentences/phrases requires a few passes. I will highlight a sample of these flaws below (there are too many to list completely in this review). I strongly recommend several  thorough, complete passes to ensure the writing is error-free, if publication in this journal is desired and if a wider readership is the goal (otherwise, the quality is just a bit too low to warrant public viewing).
(Sampled) Errors/Flaws:
Line 28: "allows to cast" should be "casts"
Line 44: "allowing training artificial agents to" should be " allow the training of artificial agents to"
Line 57:  "from their in" --> "from theirs"
Line 88:  "enabling to scale" --> "enabling the scaling of"
Line 101: "this Section" --> "this section"
Line 103: "energy, also reduce" --> "energy additionally reduces"
Line 124: "the lead to generate" --> "that lead to the generation of"
Line 153: (Add a \noindent to make sure this paragraph is left-aligned, after math equations where the text continues the thought, do not indent)
Line 180: "agent necessitates to imagine how" --> "the agent imagines"
Line 213: IMPORTANT ERROR:  "In machine learning, such a continual learning process, which requires no..." <-- this is wrong/incorrect phrase, continual learning is a very specific form of ML where agents are trained across multiple tasks without forgetting and is not exclusively unsupervised (it includes supervised, which means human annotation is needed, and unsupervised), since the writing is about "ML" and its connection to self-supervised learning, this is misleading and incorrect --> continual learning means something very specific and thus I want this phrase rewritten to be correct/accurate (ML is not "continual" to this day, it's an active area of research, including part of my specialty/expertise). Since the authors transition to the area of ML known as self-supervised/unsupervised learning, please correct the sentence to be accurate and say instead: "In some forms of machine learning that requires no..."
Line 311: "transitions to learn the model should focus" --> "transitions the model should focus on learning from"
Line 326: "is assumed able to generate" --> "is assumed to be capable of generating"
Line 335: "an high-capacity' --> "a high-capacity"
Line 344: "caring only about the preferred states to achieve" --> "caring only about the states related to accomplishing its goal(s)"
Line 356: "also shown successful for" --> "also been shown to be successful in"
Line 368: "that allows to model" --> "that facilitates the modeling of"
Line 376: "we explained" (tense incorrect) --> "we explain"
Line 376: "stodies" <-- misspelling, should be "studies"
Line 386: "how would that the agent recognizes" --> "how would the agent recognize"
Line 473: "which is" -- "which are" (plurality agreement with policies in the previous sub-phrase)
Line 406: "homeostatis" <-- misspelling, should be "homeostasis"
Equation 10 --> use \mathcal{G} for all the "G" in the equation, since its all about the same expected free energy term \mathcal{G}, otherwise, this is inconsistent mathematically and normal G needs to be defined
NOTE: this is only a sample of the errors found, please rewrite/revise the text carefully to raise the quality for the standard of this journal and for public viewing
4) Minor issue: missing citation to, on page 7, for one key discrete VAE work:
Serban, I. V., Ororbia, A. G., Pineau, J., & Courville, A. (2017, September). Piecewise Latent Variables for Neural Variational Text Processing. In Proceedings of the 2017 Conference on Empirical Methods in Natural Language Processing (pp. 422-432).
and modify the intro to something like "Bernoulli, categorical, and piecewise/piecewise-mixture distributions, " (I would consider this paper among one of the more key representative works of flexible, general discrete distributions that can be placed over the VAE prior and posterior)

Reviewer 2 Report

The work is well written and the figures well presented.

In introducing the (free energy principle) FEP however, the authors do not mention autopoiesis, which is the motivation for FEP development. This is understandable, since the authors are affiliated to what seems an engineering lab, rather than a biological or neurobiological one.

It is important that the authors incorporate and cite specifically the seminal paper of Friston (2013, Life as we know it) and the one that materializes such proposal in geochemical terms (Rubin et al, 2020). At least to determine the degree of relationship that FEP has with deep learning.

In a strict sense, perception and action in biological systems are not implementable in computed-code terms, due to the autopoietic (self-fabrication) organization that living things present. Therefore any attempt to implement FEP  only syntactically is simply a reduced simulacrum of a much larger set of a wider range of possibilities, implication and entailment.  

I would like the authors to develop this in terms of the limitations of Deep Learning itself. A guide for this can be the following paper. Ben-David, S., Hrubeš, P., Moran, S., Shpilka, A., & Yehudayoff, A. (2019). Learnability can be undecidable. Nature Machine Intelligence, 1(1), 44-48, which is much akin to what happen with the realization of autopoietic systems, thus FEP, hence learning in general.

I think that if the authors can approach this comprehensively, the article will have an even greater contribution than it has so far.

Reviewer 3 Report

This paper reviewed the state-of-the-art deep learning models for active inference. This would be useful as a starting point for machine learning practitioners to get acquainted with active inference. Some parallels between active inference and recent advances in reinforcement learning are also provided. I have several concerns that should be addressed, as listed below:

- L33: The concept of amortization is not necessarily trivial for the readers. Please provide its definition more clearly.

- L120: "Crucially, these internal states do not need to be isomorphic1 to the external ones, as their purpose is explaining sensorial states in accordance with active states, rather than memorizing the exact dynamics of the environment." Also in Fig. 2 legend, "Crucially, internal states may or may not correspond to external states, which means that hidden causes in the brain do not need to be represented in the same way as in the environment." However, if internal states are different from external states, it usually means the internal representation is suboptimal, implying that the task performance would be worse than an agent with the optimal internal representation. Please note that in Line 236, it is stated that "When one of these models is known, it would be optimal to use it." These two claims contradict each other. How much the performance is impaired when the agent employs a suboptimal internal representation seems a crucial issue, which should be discussed more carefully.

- Footnote 1: In addition, the footnote 1 states that "Beliefs of the agent about states and environment states might not be equal, but correspond to the same concepts / observations, which is adopting isomorphic structures." How is it possible to give a mathematical guarantee of the achievability of the same concepts / observations (through self-supervised or unsupervised learning)? Please cite references, which I believe the readers want to know.

- L127: "Because of these assumptions, the concept of ‘reward’ in active inference is very different from rewards in RL, as rewards are not signals used to attract trajectories, but rather sensory states that the agents aims to frequently visit in order to minimize its free energy [40]." The intended meaning of this sentence is not clear. Why can no one see the 'reward' in active inference as attracting trajectories? It is quite non-trivial why 'frequently visit' is better than 'attract trajectories' in terms of (expected) free energy minimization. Please unpack the rationale for this statement.

- L151: "Under the Free Energy Principle ...", please cite an appropriate reference that originally provides this definition and derivation. Ref. 3 is not a paper on the free-energy principle but on a more generic variational inference.

- Eq 2: The mathematical rigor of the paper seems to be weak. In Eq. 1 and elsewhere, q(\tilde{s}, \pi, \theta, \zeta) is considered, meaning that \tilde{s}, \pi, \theta and \zeta are treated as random variables in this paper. Then, the authors state that "Rewriting the last term for given policy and parameters", Eq. 2 is given as a function of \pi and \theta. However, the transformation from Eq. 1 to Eq. 2 does not hold true in general because the posterior belief q(\pi,\theta) is not necessarily equal to the prior belief p(\pi,\theta). Moreover, variational free energy is not a function of random variables but a functional of the posterior belief, or equivalently, a function of the sufficient statistics of the posterior. Additionally, why did \zeta disappear in Eq. 2? I understand that parameters are often treated as deterministic variables in the machine learning literature, so it would be allowed to consider as such. But if so, why don't the author treat parameters (and policy) as deterministic variables throughout the paper? Confusing random and deterministic variables looks not good, which undermines the credibility of this paper. Please revise.

- L159: Conditioning implies that \pi,\theta are treated as random variables. If you treat them as deterministic variables, such a conditioning should be removed from Eq. 2 for mathematical rigor.

- L240: Please provide a clear definition of 'the world model'. Does it imply stochastic differential equations? Or a statistical model in the form of a joint probability? What is the difference between the world model, and the generative model in the free-energy principle literature?

- L277: Please unpack the operation 'amortize the posterior' in this context more explicitly, which will increase the readability.

- L305: "Precision, or sensitivity [70,71], is a term used to indicate an inverse temperature." Precision can be a matrix in general while an inverse temperature is usually assumed as a single dimensional variable. So, I'm not sure if this statement is correct in general.

- L339: "However, this can be problematic, as this might lead to the model ignoring small but important features in the environment (as the pixel-wise loss is low), but waste a lot of capacity in encoding potentially irrelevant information (i.e. the exact textures of a wall)." There are a lot of papers on dimensionality reduction or feature extraction in terms of prediction ability. For example,
Lotter, W., Kreiman, G. & Cox, D. Deep predictive coding networks for video prediction and unsupervised learning. Preprint at https://arxiv.org/abs/ 1605.08104 (2016).

- L428: "This form of state matching [108] assumes that the agent knows the preferred states distribution p(s) in advance." Given the property that "these internal states do not need to be isomorphic1 to the external ones" (L120), how is it possible to anticipate the preferred states distribution p(s) in advance that corresponds to an ambiguous state representation. To the best of my knowledge, in general, the formation of such an internal representation is highly sensitive to various initial or training conditions, and the preferred p(s) must change depending on it.

- L491: "In particular, when the search over policies takes into account recursive beliefs about the future, this scheme is referred to as sophisticated inference [116]." Please explain the concept of sophisticated inference in more detail because it is relatively new for the readers.

Round 2

Reviewer 1 Report

I thank the authors for addressing my comments and believe that the additional sub-headers / text content bits/edits improve the quality of the submission. 

A few details to correct before I think the article is completely ready:
1) I would like to see the missing concerns/questions from Reviewers 1 and 2 addressed in the next iteration. Particularly, the discussion on autopoiesis, as was pointed out by Reviewer 2, needs to be corrected/explained correctly (utilizing the reference and any other related ones to bolster this bit of content). Since the article is trying to fit under the biological tag, this discussion/explanation is important to get right and to expand a bit more (as the article needs to connect AIF correctly to its biological analogues given the intended audience). I would want to see this part expanded further and detailed correctly. 
2) The citation (number 86 in the references) about the piecewise variables model (Serban et al.)  is sloppy/incorrect (the citation provided in the review provided even the citation to the conference itself not the article's arxiv preprint) -- please generate a correct citation to its published conference article -- to aid you, please use the citation bibtex provided by the formal venue here:  https://aclanthology.org/D17-1043/
3) Line 502 -- change "Parameters-driven Exploration" to "Parameter-Driven Exploration"
(overall, with the added changes, the manuscript looks fine -- please go through the document once more for final edits/checks after including/addressing the other concerns from the other 2 reviews)

Reviewer 2 Report

All the modifications the authors have made seems to be suitable, yet, I would suggest the author refer to autopoiesis a bit more carefully, by reading and directly citing Maturana and Varela (1980). Autopoiesis is not self-organization and it is not a behaivour! It is how living systems are organized and therefore why they are alive as self-producing systems. Self-production = cognition. 

So line 40 to 44 must be deleted (This self-organization process that is attributed to all living systems is called
41 autopoiesis. An example of autopoietic behavior is the morphogenesis of cells [16],
42 whose pattern formation process can be explained by the free energy principle. For more
43 complex living systems, on shorter (somatic) timescales, autopoiesis leads to the devel44
opment of the single organisms’ brains, giving rise to learning and memory functions.)

I am not sure if the arguments from line 26 to 36 are important. 

Reviewer 3 Report

I would like to thank the authors to address most of my comments. I think the revised paper can be accepted under some conditions.

I believe that the authors haven't adequately answer to the following comments that I made previously:

- "How much the performance is impaired when the agent employs a suboptimal internal representation seems a crucial issue, which should be discussed more carefully." Please discuss it more explicitly.

- The answer to my question "How is it possible to give a mathematical guarantee of the achievability of the same concepts / observations (through self-supervised or unsupervised learning)? Please cite references, which I believe the readers want to know." was provided only in the response letter "A mathematical guarantee of the achievability of the same concepts is, to our knowledge, not possible.", but not in the manuscript. Please expose that information – knowing about the limitation is quite useful for the readers.
